# Overprescribing among older people near end of life in Ireland: Evidence of prevalence and determinants from The Irish Longitudinal Study on Ageing (TILDA)

**Soraya Matthews[1], Frank Moriarty[2], Mark Ward[2], Anne Nolan[2,3], Charles Normand[1,4], Rose Anne Kenny[2], Peter May[1,2]***

**1** Centre for Health Policy and Management, Trinity College Dublin, Dublin, Ireland, **2** The Irish Longitudinal Study on Ageing, Trinity College Dublin, Dublin, Ireland, **3** Economic and Social Research Institute (ESRI), Dublin, Ireland, **4** Cicely Saunders Institute of Palliative Care, Policy & Rehabilitation, London, United Kingdom

\* peter.may@tcd.ie

**Data Availability Statement:** Both the full medications data and the EOLI data used in this analysis are not publicly available due to privacy

## Abstract

International evidence shows that people approaching end of life (EOL) have high prevalence of polypharmacy, including overprescribing. Overprescribing may have adverse side effects for mental and physical health and represents wasteful spending. Little is known about prescribing near EOL in Ireland. We aimed to describe the prevalence of two undesirable outcomes, and to identify factors associated with these outcomes: potentially questionable prescribing, and potentially inadequate prescribing, in the last year of life (LYOL). We used The Irish Longitudinal Study on Ageing, a biennial nationally representative dataset on people aged 50+ in Ireland. We analysed a sub-sample of participants with high mortality risk and categorised their self-reported medication use as potentially questionable or potentially inadequate based on previous research. We identified mortality through the national death registry (died in <365 days versus not). We used descriptive statistics to quantify prevalence of our outcomes, and we used multivariable logistic regression to identify factors associated with these outcomes. Of 525 observations, 401 (76%) had potentially inadequate and 294 (56%) potentially questionable medications. Of the 401 participants with potentially inadequate medications, 42 were in their LYOL. OF the 294 participants with potentially questionable medications, 26 were in their LYOL. One factor was significantly associated with potentially inadequate medications in LYOL: male (odds ratio (OR) 4.40, $p$ = .004) Three factors were associated with potentially questionable medications in LYOL: male (OR 3.37, $p$ = .002); three or more activities of daily living (ADLs) (OR 3.97, $p$ = .003); and outpatient hospital visits (OR 1.03, $p$ = .02). Thousands of older people die annually in Ireland with potentially inappropriate or questionable prescribing patterns. Gender differences for these outcomes are very large. Further work is needed to identify and reduce overprescribing near EOL in Ireland, particularly among men.

concerns with the small end-of-life sample. TILDA recognises the importance of transparency and reproducibility in research, and access to the full datasets including all Stata code used to generate these analyses, are available on reasonable request to the TILDA study head office. Researchers interested in using regular waves of TILDA data may access the data for free from the following sites: Irish Social Science Data Archive (ISSDA) at University College Dublin http://www.ucd.ie/issda/data/tilda/; Interuniversity Consortium for Political and Social Research (ICPSR) at the University of Michigan http://www.icpsr.umich.edu/icpsrweb/NACDA/studies/34315.

**Funding:** This work was funded by the Health Research Board (HRB) in Ireland as part of project # SDAP/2019/012 (PI: May). For more information see www.pelci.ie. The HRB website can be accessed at https://www.hrb.ie/. The funders had no role in study design, data collection and analysis, decision to publish, or preparation of the manuscript.

**Competing interests:** The authors have declared that no competing interests exist.

# 1 Introduction

## 1.1 Background

People with a limited life expectancy often have multiple health conditions that require symptom management, and thus necessitate polypharmacy [1, 2]. Polypharmacy is usually defined as taking more than five regular prescribed medications (including over the counter medications) or by the inappropriateness of the prescription [3, 4]. International evidence shows that people approaching end of life have a high prevalence of polypharmacy, including overprescribing [5–7]. While polypharmacy may be appropriate up to end of life in some cases [8], these prescribing patterns increase the risk of adverse drug reactions, hospital visits, and issues with physical and cognitive functioning [9–11]. Overprescribing can lead to increased mental and stress-related burdens on people who are near to end of life, may have adverse side effects on their physical health, and represents wasteful spending [12–15].

Deprescribing refers to the process of identifying and reducing medications that are unnecessary, or where potential harmful effects outweigh the benefits, in an effort to improve a patient's quality of life [16]. Optimising medication prescribing patterns requires clinicians to consider which medications are appropriate for each patient's circumstances, goals and life expectancy [16, 17]. Understanding potential barriers to deprescribing and patient involvement in their own medication decisions are important aspects to consider if attempting to address these issues. However, little is currently known about patterns of medication use in patients who are near the end of life in Ireland. Therefore, it is important to examining this in the context of a lack of specific policy available on deprescribing in Ireland. In order to accomplish this, it is critical that we establish the current prevalence and patterns of prescribing in Ireland before beginning to investigate the best methods to support deprescribing, where appropriate.

## 1.2 Objectives

We aimed to describe the prevalence of two undesirable outcomes, and to identify factors associated with these outcomes: potentially questionable prescribing, and potentially inadequate prescribing, in the last year of life versus not. This analysis describes for the first time the prevalence of these undesirable prescribing outcomes among older people with high mortality risk in Ireland, and identifies factors associated with these undesirable outcomes.

# 2 Methods

## 2.1 Study design & setting

We conducted secondary analysis of longitudinal cohort data. Ireland is a country of approximately 4.98 million people in north-western Europe [18]. Compared to other countries in the European Union (EU), Ireland has a relatively young population [19], meaning that the country will have proportionally larger numbers transitioning into older age and retirement within the next 20–30 years compared to the EU average [20]. Within Ireland, the number of people projected to die with an incurable disease is estimated to increase by 90% over the next 30 years [19]. This is an usually fast rate of growth; for example, neighbouring countries England and Wales [21], and Scotland [22] have projected rates of 25% and 43%, respectively. As Ireland experiences population ageing and increased longevity, multimorbidity will become more common [23].

## 2.2 Data

The Irish Longitudinal Study on Ageing (TILDA) is a biennial prospective nationally representative study of older adults residing in the community in the Republic of Ireland. It is part of

the Health and Retirement Study (HRS) family of longitudinal ageing studies [24]. Full details of the TILDA study design, participant selection and data collection are available elsewhere [25, 26]. Briefly, TILDA started in 2009–2011 (Wave 1) enrolling 8,174 participants aged 50 + and this sample is followed up every two years. All TILDA variables in this study are taken from participants' computer-assisted personal interviews (CAPI). CAPIs are conducted face-to-face in participants' place of residence and recorded on a laptop from Wave 1. All deaths in Ireland are recorded with the General Register Office (GRO), and TILDA data are linked to GRO data, in a process detailed elsewhere [27]. This allows us to identify the TILDA participants from our sample that have died in the last 365 days. Each wave of the TILDA study has been approved by the Faculty of Health Sciences Research Ethics Committee at Trinity College Dublin and all participants gave informed written consent. This paper utilised data from Wave 1 to Wave 4. All methods were performed in accordance with the relevant guidelines and regulations as per the Declaration of Helsinki.

## 2.3 Participants

We defined our sample as TILDA participants with a high mortality risk according to CAPI characteristics [28, 29]. We derived an index following prior work in the HRS family [4, 29]; this created a weighted index of demographic, health, functional, risk factors and prior service use that has high predictive power for all-cause mortality (see S1 File for summary based on sample employed in this study). The development of our full TILDA mortality risk index is discussed in detail elsewhere [30]. Participants were included in the analysis only once–at the first wave for which they exceeded the high-mortality risk threshold (11 points or above).

This sampling frame was applied to narrow down eligibility to those for whom polypharmacy and specific medicines are plausibly inappropriate or questionable. If a wider sampling frame were employed (e.g. participants with a diagnosis of any life-limiting illness), we would have included too many people for whom the medications are not obviously inappropriate or questionable, i.e. with a life expectancy where they may still reap benefits from a medication. Alternatively, if we had sampled only according to those who have already died, we would have faced two problems: first, since death is not known ex ante, any derived results are of limited policy and practice relevance; second, since TILDA interviews with the family members of deceased participants do not capture medications and many TILDA decedents do not participate in the CAPI immediately before their death, we face significant sample attrition.

## 2.4 Variables & data measurement

**2.4.1 Dependent variables.** There are two binary outcomes, (i) potentially questionable prescribing in the last year of life, and (ii) potentially inadequate prescribing in the last year of life. A systematic review on inappropriate prescribing in patients with life-limiting illness mainly identified studies where prescribing of included participants was assessed implicitly on a case-by-case basis. Other studies used measures which require extensive clinical or medication information not captured in TILDA, or which focused on a specific life-limiting condition (e.g. cancer).

In the absence of an explicit measure of inappropriate prescribing developed for people with any life-limiting illness, we defined our outcomes based on a European expert consensus panel study of medications deemed potentially inappropriate for older adults aged ≥75 years with an estimated life expectancy of ≤3 months [31]. Panellists in that study rated the extent to which prescription of different medications/classes seemed questionable, adequate or inadequate in this group, regardless of patients' underlying conditions or the drug indication. Drugs rated by at least 75% as often or always inadequate were categorised as inadequate, while of the

remaining drugs, those rated by at least 75% as questionable, or often/always questionable were categorised as questionable. Drugs/classes included in each prescribing category are listed in S1 Table in S1 File. For the purpose of this study, potentially questionable and potentially inadequate prescribing is defined as "prescribing that has potentially questionable/inadequate clinical benefit in terms of quality of care in end-of-life".

For (i), a participant has the outcome variable coded as 1 if their reported medications meet this definition of questionable prescribing in the CAPI at which they meet the eligibility criteria AND they died within 365 days of the CAPI. Otherwise the outcome is coded as 0. For (ii), a participant has the outcome variable coded as 1 if their reported medications meet this definition of inadequate prescribing in the CAPI at which they meet the eligibility criteria AND they died within 365 days of the CAPI. Otherwise the outcome is coded as 0.

**2.4.2 Independent variables.** We identified predictors in the TILDA dataset that were measured at baseline (CAPI interview at which the participant became eligible) and that we hypothesised were associated with outcome. These include: sociodemographic variables (age, gender and education); health diagnoses (cancer, heart disease, arthritis, comorbidities (e.g. more than one: heart disease, arrythmia, hypertension, diabetes, lung disease, cancer, Parkinson's, psychological illness, alcoholism, Alzheimer's, stomach ulcers or liver disease), cholesterol, incontinence, cataracts and osteoporosis); functional limitations (activities of daily living, instrumental activities of daily living and vigorous physical activity); prior healthcare use (general practitioner, emergency department, outpatient hospital and inpatient hospital visits in prior 12 months); and public health insurance (medical card and GP visit card).

In Ireland, all persons are technically entitled to receive healthcare through the public health system, but there are variations in costs, coverage and access. An alternative for residents is to pay for private health insurance for access to the public and private healthcare system [32]. A medical card is available to those receiving welfare payments, low learners and retirees and it entitles holders to receive the following free of charge: hospital care, GP visits, dental services, optical and aural services, prescription drugs and medical appliances. Those who received a slightly higher incomes are eligible for a 'GP Visit Card'. All people over age 70 and those under 70 meeting a specified low-income threshold have a medical card that subsidises prescription pharmaceuticals. Those without a medical card pay capped co-payments for prescriptions. A small proportion of people over 70 prefer not to take up a medical card as it restricts some of their choice in accessing certain types of health care; this group is wealthier and healthier than the population average [33].

## 2.5 Bias & missing data

With respect to external validity, bias concerns are very low. TILDA Wave 1 was sampled in a sophisticated way to represent the population of interest [34]. With respect to internal validity, the biggest concern was missing values, which can arise if a participant refused to respond to a question or did not know the relevant answer. Where a participant was missing data on any of our independent variables we imputed with that participant's response from the most recent preceding wave in which they did respond. We checked the robustness of our results by repeating primary analyses without those observations for whom predictors were imputed.

## 2.6 Statistical methods

We reported the prevalence of these undesirable outcomes in the sample using descriptive statistics. We conducted multivariable logistic regression to identify factors that are significantly associated with outcome, expressing relationships in an odds ratio (OR).

# 3 Results

## 3.1 Characteristics of the sample

The identification of eligible TILDA participants is provided in S2 Table in S1 File. There are 525 participants included in the analysis.

## 3.2 Descriptive data

The sample is presented according to included baseline variables in Table 1. Just over half of the sample were male (53.1%) and a majority (58.5%) achieved no higher than primary education. A large majority of the sample were aged 75 or older (81.5%). Over three quarters of the sample had a heart risk factor (at least one diagnosis of: hypertension, arrhythmia, high cholesterol) (82.5%) and more than one serious chronic condition (75.8%). On average over 12 months, participants visited the GP 9.3 times, visited hospital as an outpatient 3.7 times and spent 1.21 nights as an inpatient in hospital. Our sample demonstrated that 94% of people have a medical card, reflecting the high age profile of the sample.

## 3.3 Outcome data

The outcome variables are presented in Table 2. Of the 525 eligible participants, 401 (76.4%) had potentially inadequate medications and 294 (56%) had potentially questionable

**Table 1. Descriptive data of total sample ($n$ = 525).**

| Variables | N | % |
|---|---|---|
| **Gender (male)** | 279 | 53.1% |
| **Education** | | |
| *Primary/never* | 307 | 58.5% |
| *Secondary* | 140 | 26.7% |
| *Third/higher* | 78 | 14.9% |
| **Age group** | | |
| *Under 70 years* | 25 | 4.8% |
| *70–74 years* | 72 | 13.7% |
| *75–79 years* | 79 | 15.0% |
| *80–84 years* | 140 | 26.7% |
| *85 years and over* | 209 | 39.8% |
| **Cancer** | 190 | 36.20% |
| Serious heart condition* | 198 | 37.7% |
| **Heart risk factor(s)§** | 433 | 82.45 |
| **More than one serious chronic condition** | 398 | 75.8% |
| **Activities of daily living** | | |
| *1 ADL* | 71 | 13.5% |
| *2 ADL* | 53 | 10.1% |
| *3 or more ADL* | 189 | 36.0% |
| **Medical card or GP visit card** | 491 | 93.5% |
| | **Mean** | **SD** |
| **GP visits (12 months)** | 9.33 | 14.0 |
| **Outpatient hospital visits (12 months)** | 3.70 | 9.8 |
| **Inpatient/emergency dept hospital visits (12 months)** | 1.21 | 2.7 |

*At least one diagnosis of: congestive heart failure, heart attack, stroke, angina. §At least one diagnosis of: hypertension, arrhythmia, high cholesterol.

**Table 2. Outcome data in the analytic sample (*n* = 525).**

| At wave of eligibility | Sample | Potentially inadequate medications? | | Potentially questionable medications? | |
|---|---|---|---|---|---|
| *In last year of life?* | | *No* | *Yes* | *No* | *Yes* |
| *No* | 457 (87%) | 98 | 359 | 189 | 268 |
| *Yes* | 68 (13%) | 26 | 42 | 42 | 26 |
| **Total** | **525** | **124** | **401** | **231** | **294** |

medications. Sixty-eight participants were in the last year of life, of whom 42 (61.8%) had potentially inadequate medications and 26 (38.2%) had potentially questionable medications. The most prevalent potentially questionable medications were antiplatelet and anticoagulant drugs, and antihypertensives (see S5 Table in S1 File). Lipid-lowering agents and medications for the treatment of osteoporosis (e.g. calcium and vitamin D, bisphosphonates) were the most common potentially inadequate medications (S6 Table in S1 File). Our analysis showed that those with a medical card had poor outcomes, reflecting the higher age and higher illness burden of that group, but this does not imply any causal relationship (Table 3).

## 3.4 Primary analyses

Regression results for potentially inadequate medications in LYOL are presented in Table 3, with statistically significant (p<0.05) associations highlighted in bold. There is one such predictor: male (OR 4.40, p = .004).

**Table 3. Multivariable regression results for potentially inadequate medications.**

| Variables | Odds Ratio | P>z | [95% Conf. | Interval] |
|---|---|---|---|---|
| **Male** | **4.40** | **<0.01** | **0.14** | **18.51** |
| **Education** | | | | |
| *Secondary* | 0.38 | 0.13 | 0.11 | 1.35 |
| *Third/higher* | 0.61 | 0.42 | 0.18 | 2.04 |
| **Age group** | | | | |
| *70–74 years* | 1.66 | 0.68 | 0.15 | 18.51 |
| *75–79 years* | 2.57 | 0.42 | 0.26 | 25.90 |
| *80–84 years* | 1.44 | 0.76 | 0.14 | 15.47 |
| *85 years and over* | 4.89 | 0.17 | 0.52 | 46.08 |
| **Cancer** | 1.07 | 0.90 | 0.37 | 3.07 |
| **Any serious heart condition** | 0.63 | 0.39 | 0.22 | 1.80 |
| **Any heart risk factor(s)** | 1.44 | 0.61 | 0.36 | 5.73 |
| **More than one serious chronic condition** | 1.29 | 0.11 | 0.76 | 1.36 |
| **Activities of daily living** | | | | |
| *1 ADL* | 0.34 | 0.33 | 0.04 | 3.06 |
| *2 ADL* | 1.66 | 0.49 | 0.39 | 7.03 |
| *3 or more ADL* | 2.87 | 0.05 | 1.00 | 8.27 |
| **GP visits (12 months)** | 0.98 | 0.41 | 0.94 | 1.02 |
| **Outpatient hospital visits (12 months)** | 1.02 | 0.08 | 1.00 | 1.05 |
| **Inpatient/emergency dept hospital visits (12 months)** | 1.03 | 0.67 | 0.91 | 1.16 |
| **Medical card or GP visit card** | - | - | - | - |

note: Medical card dropped as all participants with potentially inadequate medications had a medical or GP visit card.

**Table 4. Multivariable regression results for potentially questionable medications.**

| Variables | Odds Ratio | P>z | [95% Conf. | Interval] |
|---|---|---|---|---|
| **Gender (male)** | **3.37** | **<0.01** | **1.55** | **7.33** |
| **Education** | | | | |
| *Secondary* | 0.74 | 0.49 | 0.31 | 1.75 |
| *Third/higher* | 0.65 | 0.44 | 0.22 | 1.94 |
| **Age group** | | | | |
| *70–74 years* | 1.52 | 0.73 | 0.14 | 16.74 |
| *75–79 years* | 4.14 | 0.21 | 0.44 | 38.99 |
| *80–84 years* | 1.57 | 0.70 | 0.16 | 15.88 |
| *85 years and over* | 8.06 | 0.06 | 0.90 | 72.13 |
| **Cancer** | 0.94 | 0.89 | 0.38 | 2.31 |
| **Any serious heart condition** | 0.64 | 0.32 | 0.27 | 1.53 |
| **Any heart risk factors** | 1.64 | 0.36 | 0.57 | 4.67 |
| **Any serious chronic condition** | 1.09 | 0.53 | 0.83 | 1.44 |
| **Activities of daily living** | | | | |
| *1 ADL* | 0.48 | 0.36 | 0.10 | 2.34 |
| *2 ADL* | 1.99 | 0.25 | 0.61 | 6.48 |
| *3 or more ADL* | **3.97** | **<0.01** | **1.61** | **9.82** |
| **GP visits (12 months)** | 0.98 | 0.29 | 0.94 | 1.02 |
| **Outpatient hospital visits (12 months)** | **1.03** | **0.02** | **1.00** | **1.05** |
| **Inpatient/emergency dept hospital visits (12 months)** | 1.00 | 0.95 | 0.87 | 1.14 |
| **Medical or GP visit card** | 2.73 | 0.36 | 0.32 | 23.08 |

Regression results for potentially questionable medications in LYOL are presented in Table 4, with statistically significant ($p<0.05$) associations highlighted in bold. There are three such predictors: male (OR 3.37, $p$ = .002); three or more ADLs (OR 3.97, $p$ = .003); and number of outpatient hospital visits (OR 1.03, $p$ = .02).

## 3.5 Other analyses

We conducted sensitivity analyses without those observations where predictors were missing and imputed. These results are reported in S3-S6 Tables in S1 File; our main findings and interpretation are unaffected. Post-hoc analysis of prescribing differences between males and females identified that antiplatelet and anticoagulant medications, and lipid-lowering agents were more prevalent among male participants, while osteoporosis-related medications were more prevalent among female participants (S5 and S6 Tables in S1 File).

# 4 Discussion

## 4.1 Key results and interpretation

In a nationally representative sample of 525 older people with high mortality risk in Ireland, 401 (76%) had potentially inadequate and 294 (56%) potentially questionable medications. Prevalence of these prescribing patterns among those in LYOL was 62% and 38% respectively. One factor was significantly associated with potentially inadequate medications in LYOL: male (OR 4.40, $p$ = .004) Three factors were associated with potentially questionable medications in LYOL: male (OR 3.37, $p$ = .002); three or more activities of daily living (ADLs) (OR 3.97, $p$ = .003); and outpatient hospital visits (OR 1.03, $p$ = .02).

Comparatively lower prevalence of the specified prescribing patterns in LYOL is encouraging to the extent that they may indicate a level of deprescribing is occurring as people approach end of life. However, approximately 30,000 people die annually in Ireland (the majority of them over the age of 50), and our data suggest that a large proportion of these are at risk of inappropriate polypharmacy.

Tackling high prevalence of overprescribing near end of life requires ex ante identification of those at greatest risk. Multivariable regression results (Tables 3 and 4) indicate that male sex is the strongest predictor of both undesirable outcomes. The odds of men receiving potentially inadequate medications were 4.4 times higher (than the odds of women receiving potentially inadequate medications), after controlling for other important predictors of prescribing. Similarly, the odds of men receiving potentially questionable medications were 3.4 times higher (than the odds of women receiving questionable medications), after controlling for the same predictors. Three or more ADL difficulties and outpatient engagements were also associated with potentially questionable medications, but neither cancer nor cardiovascular disease were associated with outcome.

Examining the prevalence of potentially inadequate and questionable medications among male and female participants, differences were noted across medications used in the prevention and treatment of cardiovascular disease (antiplatelet and anticoagulant drugs, lipid-lowering drugs), with higher prevalence among males, and to a lesser extent, higher prevalence of medication for osteoporosis among females. These patterns are reflective of the patterns of prescribing observed in middle-aged and older adults more generally, with noted undertreatment of women compared to men given the same level of cardiovascular morbidity [35, 36], while older men with similar risk of fracture to women are less likely to be treated for osteoporosis [37]. This suggests that being of high mortality risk does little to reduce the disparity in prescribing between sexes. There were also high rates of medical or GP visit card eligibility (based on income) among those with questionable or inadequate medications, potentially reflecting an impact of higher burden and severity of co-morbidities, or greater access to care.

Much of the existing literature on inappropriate prescribing among people with limited life expectancy has focussed on use of preventive medications [38], where the likely benefits will not be realised within a patient's life expectancy [39]. Most studies have focussed on people with a specific life-limiting condition, most often cancer. The most often cited preventive medications prescribed inappropriately were lipid-lowering drugs and antihypertensives, similar to the findings of our study, as well as antidiabetic drugs, which had a relatively low prevalence in the current study [38]. Often such medications are only stopped in the last month of life [40]. This underlines the importance of reviewing and discussing the need and benefit of such long-term medications to align treatment with patients' goals, in general but particularly among people with life-limiting illness. It is also worth noting that while overprescribing is the focus of much research on optimal medication use near end of life, under prescribing is also an important issue, where use of symptom-relieving medications that enhance quality of life may be insufficient [40]. Care for such patients should focus not just on reducing medications, but ensure therapy maximises quality and quantity of life and is aligned with patient priorities [39].

Ireland is ranked fourth worldwide for end-of-life care, meaning that there is relatively high per capita palliative care provision at the population level [41, 42]. Our paper highlights that even in a setting with well-established palliative care services, the risk of inappropriate prescribing among people with limited life expectancy in significant. However, these deficiencies in patient experience must be understood in the context of other challenges; previous research in TILDA demonstrated strong evidence for undermanagement of illness symptoms near end of life, whereas overprescribing is a form of overtreatment [43].

### 4.2 Limitations

TILDA CAPI data, including both our predictors and our medications for outcome measures, are self-report, which carries risk of bias from measurement and memory error. We considered a large number of potential predictors from TILDA's rich individual-level dataset, but it is possible that an important unobserved factor associated both with sex and prescribing patterns is missing. We were only able to identify one set of explicit criteria to identify suboptimal prescribing for which sufficient clinical information was available in TILDA and not focusing on a group with a specific life-limiting illness. Although these criteria were focussed on those aged ≥75 years with an estimated life expectancy of ≤3 months, a very high proportion of our sample (81.5%) were aged 75 years and over. Data restrictions mean that we quantify those with potentially inappropriate prescribing patterns in the last year of life, but we cannot identify the extent to which these patterns are discontinued in the end-of-life phase. Potentially patterns of prescribing near end of life are changing over time, but the TILDA sample at each wave is too small to analyse rigorously such trends. Similarly, temporal comparison of men and women's patterns before and during LYOL will require larger datasets–and our analyses highlight the need for investigating sex differences in this population.

## 5 Conclusion

Thousands of older people die annually in Ireland with potentially inappropriate or questionable prescribing patterns. Gender differences for these outcomes are very large. Further work is needed to identify and reduce overprescribing near EOL in Ireland, particularly among men.

## Supporting information

**S1 File.**
(DOCX)

## Author Contributions

**Conceptualization:** Soraya Matthews, Frank Moriarty, Peter May.

**Data curation:** Soraya Matthews, Frank Moriarty, Mark Ward, Anne Nolan, Charles Normand, Rose Anne Kenny, Peter May.

**Formal analysis:** Soraya Matthews, Frank Moriarty, Anne Nolan, Charles Normand, Rose Anne Kenny, Peter May.

**Funding acquisition:** Peter May.

**Investigation:** Soraya Matthews, Frank Moriarty.

**Methodology:** Soraya Matthews, Frank Moriarty, Peter May.

**Visualization:** Soraya Matthews.

**Writing – original draft:** Soraya Matthews, Frank Moriarty, Mark Ward, Anne Nolan, Charles Normand, Rose Anne Kenny, Peter May.

**Writing – review & editing:** Soraya Matthews, Frank Moriarty, Mark Ward, Anne Nolan, Charles Normand, Rose Anne Kenny, Peter May.

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
