## [Decision Letter · Decision Letter 0]

4 Aug 2022

PONE-D-22-18482Overprescribing among older people near end of life in Ireland: evidence of prevalence and determinants from The Irish Longitudinal Study on Ageing (TILDA)PLOS ONE

Dear Dr. May,

Thank you for submitting your manuscript to PLOS ONE. After careful consideration, we feel that it has merit but does not fully meet PLOS ONE’s publication criteria as it currently stands. Therefore, we invite you to submit a revised version of the manuscript that addresses the points raised during the review process.

Besides the comments raised by the reviewers, I consider that several points need to be addressed. I have ordered them according to their relevance:

1- In Objectives, Discussion and other parts of the manuscript, it is stated that the paper describes overprescription in the last year of life. However, this is misleading. In fact, only 13% of the study subjects were in her/his last year of life. This should be amended in the whole text. I would stress the description of the population in the Abstract. There is no definition of the inclusion criteria, but a vague sentence about “We identified mortality through the national death registry”.

2- Public health insurance. According to Table 1, 34 participants (6,5 %) were no holders of Medical Card. For an international audience, it would be helpful a brief description of the Public health insurance schemes in Ireland and the criteria of residents to apply for them. A very remarkable finding is that 401 out 491 (82%) Medical Card holders received potentially inadequate medications, but none of non-holders (Tables 1 and 3). This fact should be developed and explained. I wonder if it suggests any lack of access to prescriptions in no card holders.

3- As the reviewers highlight, it would be would be helpful to include further information about the “Mortality Index” and specifically describe the items employed for such score. It seems plausible that the predictors employed are related to morbidity and, consequently, to the need of prescriptions.

4- The authors count with a database that includes information of 4 different survey waves. I wonder if it would be possible to observe the temporal evolution of prescription patterns.

5- In Methods, it is explained that if follow-up confirmed that a participant in the study had died, a family member or friend was asked to complete an interview (EOLI). If I fully understand it, the information collected in EOLI was not employed in the study, so the reference to EOLI could be avoided.

6- Pag 8, line 175: the expression ”comorbidities” is too ambiguous and needs a better description.

7- There is a typo in Table 3 (95% Conf. Interval for Males)

We look forward to receiving your revised manuscript.

Kind regards,

Juan F. Orueta, MD, PhD

Academic Editor

PLOS ONE

Journal Requirements:

"This work was funded by the Health Research Board in Ireland as part of project # SDAP/2019/012 (PI: May). For more information see www.pelci.ie."

"This work was funded by the Health Research Board (HRB) in Ireland as part of project # SDAP/2019/012 (PI: May). For more information see www.pelci.ie. The HRB website can be accessed at https://www.hrb.ie/. The funders had no role in study design, data collection and analysis, decision to publish, or preparation of the manuscript."

Reviewers' comments:

Reviewer's Responses to Questions

**Comments to the Author**

1. Is the manuscript technically sound, and do the data support the conclusions?

Reviewer #1: Partly

Reviewer #2: Yes

2. Has the statistical analysis been performed appropriately and rigorously? 

Reviewer #1: Yes

Reviewer #2: Yes

3. Have the authors made all data underlying the findings in their manuscript fully available?

Reviewer #1: Yes

Reviewer #2: Yes

4. Is the manuscript presented in an intelligible fashion and written in standard English?

Reviewer #1: Yes

Reviewer #2: Yes

5. Review Comments to the Author

Reviewer #1: Thank you for the opportunity to review this manuscript, titled "Overprescribing among older people near end of life in Ireland: evidence of prevalence and determinants from The Irish Longitudinal Study on Ageing (TILDA)." This study uses TILDA to investigate potential over and under prescribing patterns in the last year of life. The authors identify the cohort (or study sample) using a predictive mortality index, investigate their prescriptions at the time they meet the 'likely to die' threshold, and follow until i) death or ii) for one year. Providing quality end of life care is important, and describing current practice is essential to make improvements. Overall, I found this manuscript a little confusing and the sample was on the smaller side, especially with increasing granularity. 

I have some comments for the authors to consider:

1. It would be helpful for readers if the index used was reproduced in supplementary material. For example, diabetes and lung disease were included in the index, and oral antidiabetics, excluding metformin, and systemic drugs for obstructive airway diseases are included in the questionable drugs classes. So, it seems the cohort chosen may have been more likely to be using a 'questionable drug' (i.e. overestimates inappropriate use).

2. The interpretation of the OR in section 4.1 appears to be incorrect. I don't believe that either the outcome of potentially inadequate or questionable medications was rare enough to justify approximating OR for RR (in which case, saying 4.4x more likely would be correct). So, something like 'The odds of [men] receiving potentially inadequate medications were 4.4 times higher [than the odds of women receiving potentially inadequate medications], after controlling for other important predictors of prescribing' would be more accurate.

3. I understand you are using a previous classification, but taken without the context of that study, the terms 'potentially inadequate' and 'potentially questionable' are not as clear as, say, underprescribing or potentially inappropriate prescribing. For example, 'potentially inadequate' could be interpreted as underprescribing (as adequate can be quality or quantity'), but in this case it means '[prescribing that has] potentially inadequate clinical benefit' (which qualifies, it means quality). It may be helpful for a global audience to make sure this is clear.

4. The paragraph beginning on page 14, line 257 - could you/did you look at the differences in trends between men and women in LYOL vs not? That might be a more interesting inclusion that the current paragraph, which is a little jarring as the literature cited talks about undertreatment of drugs when those same drugs are on the questionable and inadequate lists (i.e. in general, it would be better quality prescribing if people weren't on them at end of life).

5. Do you have any sense as to how well the index worked? For your sample, would prescribers have known they were prescribing at end of life, so that could factor into decisions to continue or cease medicines?

6. Did you have any information on family support that could be used in the regression?

Finally, the manuscript could do with a good proofread e.g. page 3 line 60, page 4 line 89, page 14 line 246. Table 3 doesn't appear to have bolded all statistically significant results.

Reviewer #2: Overprescribing among older people near end of life in Ireland: evidence of prevalence and determinants from The Irish Longitudinal Study on Ageing (TILDA) is an interesting and well written manusript. I have some minor comments for the authors:

1. line 175 - could you please be more precise about the health diagnoses - ie why is cholesterol per se in the bracket, which comorbidities do you mean etc

2. line 201 - heart risk factor should be defined more precisely in the way it was presented in the legent of Table 1.

3. line 204 - did the patients have more than one serious condition or more than one health condition. There seem to be a discrepancy between the text and the table (1).

6. PLOS authors have the option to publish the peer review history of their article (what does this mean?). If published, this will include your full peer review and any attached files.

Reviewer #1: No

Reviewer #2: No

---

## [Author Response · Author response to Decision Letter 0]

26 Aug 2022

We would like to thank the editor and reviewers for their comments. We have uploaded a document with our responses. Thank you for your consideration.

---

## [Decision Letter · Decision Letter 1]

20 Sep 2022

PONE-D-22-18482R1Overprescribing among older people near end of life in Ireland: evidence of prevalence and determinants from The Irish Longitudinal Study on Ageing (TILDA)PLOS ONE

Dear Dr. May,

Thank you for submitting your manuscript to PLOS ONE. After careful consideration, we feel that it has merit but does not fully meet PLOS ONE’s publication criteria as it currently stands. Therefore, we invite you to submit a revised version of the manuscript that addresses the points raised during the review process.

The authors have been responsive to several of the previous comments. However, I consider that some major points need to be addressed.

1.- Population of study:

If I have interpreted the manuscript correctly, the population of study consists in 525 individuals approaching end of life (score of 11+ points on the TILDA mortality index). If my inference is correct, the text following line 186 is misleading. Were the two outcome variables coded as 1 exclusively in participants when their reported medications meet the defined criteria AND they died within 365 days of the CAPI? Please clarify.Moreover, revise the objectives (line 107 and following). This section includes two expressions that are close, but different. The first sentence states: “We aimed to describe… in the last year of life versus not”, but the second one: “This analysis quantifies … among older people with high mortality risk in Ireland”Lines 58, 60 and 267. If my interpretation is correct, the expression “LYOL” should be replaced by “approaching end of life” or another similar.Besides, I will suggest a modification to the first paragraph of Discussion section. Usually, the first sentence summarizes the major findings related to the objectives of the study. The comparison of percentages in people in their last year of life and not is of interest and could be included later in this section, but not in the first sentence. It does not provide a clear answer to the objectives of the study.

2.- Public health insurance

According to the manuscript, the inclusion in the Medical Card Program is associated with potentially questionable medications. However, such association is even more pronounced for inadequate medications, since all of the individuals receiving inadequate medications were Medical Card holders and none of the no-holders received inadequate medications. I agree that it is not possible to estimate an OR when one group includes zero subjects, but the association between Medical Card and inadequate medications has to be recognized and developed.Besides, the relationship between Medical Card and potentially inadequate and questionable medications needs further explanation in the Discussion section. Did the no-holders receive a better attention due to their lack of questionable medications? Otherwise, does this finding suggest any lack of access to prescriptions in no card holders? Please, provide plausible hypotheses for this fact.

Minor comment

Line 158: TILDA EOLI is not included in the manuscript, so this sentence may be difficult to be interpreted.

We look forward to receiving your revised manuscript.

Kind regards,

Juan F. Orueta, MD, PhD

Academic Editor

PLOS ONE

Reviewers' comments:

Reviewer's Responses to Questions

**Comments to the Author**

1. If the authors have adequately addressed your comments raised in a previous round of review and you feel that this manuscript is now acceptable for publication, you may indicate that here to bypass the “Comments to the Author” section, enter your conflict of interest statement in the “Confidential to Editor” section, and submit your "Accept" recommendation.

Reviewer #1: All comments have been addressed

Reviewer #2: All comments have been addressed

2. Is the manuscript technically sound, and do the data support the conclusions?

Reviewer #1: Yes

Reviewer #2: Yes

3. Has the statistical analysis been performed appropriately and rigorously? 

Reviewer #1: Yes

Reviewer #2: Yes

4. Have the authors made all data underlying the findings in their manuscript fully available?

Reviewer #1: Yes

Reviewer #2: Yes

5. Is the manuscript presented in an intelligible fashion and written in standard English?

Reviewer #1: Yes

Reviewer #2: Yes

6. Review Comments to the Author

Reviewer #1: (No Response)

Reviewer #2: The manuscript is interesting and well written. All previous comments have been addressed I have no further.

7. PLOS authors have the option to publish the peer review history of their article (what does this mean?). If published, this will include your full peer review and any attached files.

Reviewer #1: No

Reviewer #2: No

---

## [Editor Report · Decision Letter 2]

9 Nov 2022

Overprescribing among older people near end of life in Ireland: evidence of prevalence and determinants from The Irish Longitudinal Study on Ageing (TILDA)

PONE-D-22-18482R2

Dear Dr. May,

We’re pleased to inform you that your manuscript has been judged scientifically suitable for publication and will be formally accepted for publication once it meets all outstanding technical requirements.

Kind regards,

Juan F. Orueta, MD, PhD

Academic Editor

PLOS ONE
---

## [Editor Report · Acceptance letter]

18 Nov 2022

PONE-D-22-18482R2 

Overprescribing among older people near end of life in Ireland: evidence of prevalence and determinants from The Irish Longitudinal Study on Ageing (TILDA) 

Dear Dr. May:

I'm pleased to inform you that your manuscript has been deemed suitable for publication in PLOS ONE. Congratulations! Your manuscript is now with our production department. 

Kind regards, 

on behalf of

Dr. Juan F. Orueta 

Academic Editor

PLOS ONE